# Meta-Knowledge and Multi-Task Learning-Based Multi-Scene Adaptive Crowd Counting

**DOI:** 10.3390/s22093320

**Published:** 2022-04-26

**Authors:** Siqi Tang, Zhisong Pan, Guyu Hu, Yang Wu, Yunbo Li

**Affiliations:** 1Control Engineering College, Army Engineering University of PLA, Nanjing 210007, China; tangsiqi3036@163.com (S.T.); huguyu@189.cn (G.H.); 18252059269@163.com (Y.L.); 2Beijing Information and Communications Technology Research Center, Beijing 100036, China; 13218082261@163.com

**Keywords:** multi-scene adaptive, crowd counting, meta-knowledge, multi-task learning

## Abstract

In this paper, we propose a multi-scene adaptive crowd counting method based on meta-knowledge and multi-task learning. In practice, surveillance cameras are stationarily deployed in various scenes. Considering the extensibility of a surveillance system, the ideal crowd counting method should have a strong generalization capability to be deployed in unknown scenes. On the other hand, given the diversity of scenes, it should also effectively suit each scene for better performance. These two objectives are contradictory, so we propose a coarse-to-fine pipeline including meta-knowledge network and multi-task learning. Specifically, at the coarse-grained stage, we propose a generic two-stream network for all existing scenes to encode meta-knowledge especially inter-frame temporal knowledge. At the fine-grained stage, the regression of the crowd density map to the overall number of people in each scene is considered a homogeneous subtask in a multi-task framework. A robust multi-task learning algorithm is applied to effectively learn scene-specific regression parameters for existing and new scenes, which further improve the accuracy of each specific scenes. Taking advantage of multi-task learning, the proposed method can be deployed to multiple new scenes without duplicated model training. Compared with two representative methods, namely AMSNet and MAML-counting, the proposed method reduces the MAE by 10.29% and 13.48%, respectively.

## 1. Introduction

Recent years have witnessed the occurrence of extensively crowded scenes in public places such as walkways, parks, sport events, concerts and holiday parades, which poses major threats to public security [1]. As crowd density is one of the major descriptions of the crowd’s security status, surveillance-based crowd counting has attracted much attention in machine learning and computer vision. The current methods mainly consider crowd counting as a standard supervised learning problem [2]. The common way is to place the collected training images with pedestrian location annotations in a data pool. Afterwards, a convolutional neural network (CNN) is used to train the shadowing from the image to the density map. In the past few years, multi-scale information fusion [3,4,5], Attention mechanism [6,7] and Multi-task Learning [8,9] have been proposed for crowd counts, which has achieved remarkable progress.

The supervised single-image crowd counting method [3,4,5,6,7,8,9] achieves a promising performance on the standard dataset. However, the following limitations still exist in real-world surveillance applications.

Surveillance cameras often need to be installed in new positions. Due to the differences in lighting, background, camera positions and camera angles between scenes, the crowd counting method with supervised learning is difficult to adjust to new scenes effectively. High-performance generic crowd density estimation models have been demonstrated with significantly reduced performance when tested on other datasets. It severely limits the application of supervised crowd counting methods in new scenes.The domain features of practical surveillance applications are ignored. Unlike image classification and detection in the field of computer vision, a crowd counting model is not required to accurately process arbitrary single images in real-world surveillance applications. In contrast, because the positions and angles of each camera are stationary, local models that are more adapted to each camera deployment scene tend to be more accurate than generic models trained on all training data pools.

The above deployment problem is an essential issue to be addressed in real-world surveillance applications, where the goal is to obtain a density estimation model applicable to new scenes using a small amount of labeled data. It has been defined as a scene adaptive counting problem in studies [10,11]. In existing studies, the finetune mechanism [10], adversarial training [12,13,14] and meta-learning [11] were adopted to effectively reduce the training data and computational cost for inter-scene transfer of models. However, the above model transfer mechanism has the following limitations:The meta-knowledge analysis of the crowd density estimation problem is not comprehensive. Most studies focus on crowd features, multi-scale target recognition, foreground segmentation and occlusion analysis for a single image, while the common knowledge of inter-frame, shared by the crowd image in each scene, has been neglected. Due to the stationary camera position, the surveillance video frames in each scene often exhibit a change in the foreground crowd, without any change in the background. This time-domain change provides important prior information: the region that remains stable in the time domain between two frames is more likely to be background; the region where change occurs between frames of the same scene is likely to be the foreground crowd. Most of the existing studies adopt a single image as network input, thus making it difficult to learn the time-domain knowledge, which degrades the generalization ability of the model.In practical surveillance applications, it is often necessary to install multiple new surveillance cameras simultaneously, which involves the deployment of multiple new scenes. However, most of the above studies with transfer can merely deploy to one new scene each time by adapting the model from the source domain to the target domain. When deploying in multiple new scenes, such methods need to train the transfer process for multiple times, bringing a large workload for the staff.

Considering the above limitations, this paper proposes a coarse-to-fine pipeline for multi-scene adaption problem of real-world surveillance applications. In the coarse-grained process, the meta-knowledge of all scenes is analyzed and then adopted to optimize a generic density regression network structure. On this basis, considering the fine-grained differences between scenes, the overall counting regression from estimated density map for each known and unknown scene is considered a homogeneous subtask. Through a robust multi-task learning method, the regression parameters of each scene are explicitly learned, which are suitable for various scenes.

The contributions of the proposed coarse-to-fine method in this paper are listed below:The knowledge in crowd density estimation is comprehensively discussed. The scene-shared knowledge, which is generic among all scenes, is defined as meta-knowledge, including crowd features, background features, multi-scale knowledge in the spatial domain and inter-frame knowledge in the time domain. Such meta-knowledge plays the most crucial role in a model’s generalization capacity. Existing methods often subconsciously or implicitly employ several meta-knowledge, whereas this paper firstly analyzes such knowledge in a formal way.To the best of our knowledge, the meta-knowledge of the inter-frame temporal change is firstly considered in the field of crowd counting. Different from existing methods that focus on single-image crowd counting, a two-stream network is proposed in this paper. Leveraging difference mechanism of the high-level features, the two-stream network can learn the difference between video frames in the same scene, which improves segmentation between crowd foreground and static background. Since the perception capability of this network comes from the encoding of scene-independent meta-knowledge, it has favorable generalization to various new scenes.Unlike finetune [10,11] or adversarial training mechanisms [12,13,14], which require training each model’s parameter to adapt to a specific new scene, we apply a robust multi-task learning method to regress the person count of all scenes from an estimated density map simultaneously. Through multi-task learning, the commonality and difference between each subtask can be obtained. As a result, the regression parameters suitable for multiple unknown scenes can be obtained with a small amount of data, effectively saving the training cost required for deployment in a real-world surveillance application.

## 2. Related Work

### 2.1. Crowd Counting

Aiming to estimate the number of people, crowd counting methods can be divided into three main categories: single target location, direct regression and density map regression. Single target location-based methods aim to locate and then count each person by sliding window pedestrian detection, segmentation or tracking. Due to crowded occlusion, their performance degrades with the increase of crowd density. Direct regression methods, including foreground segmentation, feature extraction and counting regression, ignore the crowd’s spatial distribution information and thus can barely have satisfying performance.

To reduce the above problems, Lempitsky et al. [15] introduce density map regression methods that learn a mapping between local features and corresponding density maps. In recent years, benefiting from the powerful non-linear mapping capacity and powerful feature representation of CNNs [16], density map regression approaches based on CNN obtain significant improvement and show promising performance. Cao et al. [17] adopt scale aggregation modules to extract multi-scale features and propose a novel training loss combining Euclidean loss and local pattern consistency loss. To improve generalization capability, Shi et al. propose decorrelated ConvNet [18], where a pool of decorrelated regressors are trained. Considering the fact that detection mechanism is more suitable to low density scenes, while regression is more applicable for congested areas, Liu et al. propose DecideNet [19], which can adaptively decide whether to adopt regression pipeline or detection pipeline for different locations based on its density conditions.

### 2.2. Scene Adaptive Crowd Counting

Despite the remarkable progress of the aforementioned supervised counting methods achieved in labeled datasets, severe performance degradation is usually observed when deploying the trained model in new scenes of the surveillance application. This is mainly caused by the domain difference between the training and deployment, which can be listed as follows.

Different background. Background regions (have no person instances) including buildings, trees and other confusing objects, may vary dramatically in different crowd scenes. For example, the background of frames collected in a park may contain more green background elements such as grasses and trees, while crowded frames of a street contain more gray areas such as buildings and streets. As background usually has a similar appearance or colors with the crowd, it is necessary to learn the background knowledge to improve the accuracy of the model in a specific scene.Scale variation. It is the primary problem in the field of density estimation, as the scales of objects (such as the sizes of people’s heads) vary according to their distance from the camera [20]. Owing to the different locations and angles of cameras, scale distributions often vary substantially among scenes.Crowd distribution. As the surveillance camera is stationary, each scene has some specific areas such as walls, trees and sky area that are rarely positioned at the same place as a person and the positions of these areas are usually different among scenes. Diverse crowd densities and distributions in different scenes reduce the accuracy of crowd density estimation.

With the aforementioned challenges faced by real-world surveillance applications among various scenes, domain adaptive crowd counting has aroused the interest of more and more researchers. Existing works of this field can be grouped into three categories.

#### 2.2.1. Arbitrary Scene Methods

The arbitrary scene (or all scenes) methods simply pool training images of all scenes and train an overall model. To promote the scene adaptive capacity of the CNN network, Chen et al. [21] propose the novel Variational Attention technique for explicitly modeling the attention distributions for different domains. Considering scale variations and complex scenes, Wei et al. [22] apply Transformer backbone to learn scale-adaptive feature representations. Moreover, Yan et al. [23] adopt channel attention to guide the extraction of domain-specific feature representation and thus tackle the variations in scene contexts, crowd densities and head scales.

The core of the above methods is training a model that works well for diverse backgrounds, crowd distributions and scales. However, in practical applications with multiple scenes, learning a generic model that works well in all scenes is suboptimal compared to learning and deploying a model that is specialized for a specific scene.

#### 2.2.2. Domain Adaptation-Based Methods

Domain adaptation-based crowd counting aims to learn domain-invariant feature representations. Methods along this line can be generally categorized into two types: Criterion-based methods and adversarial training-based methods.

Criterion-based methods aim to reduce the distribution variance between two scenes. To address the scale difference across scenes and datasets, Ma et al. [24] propose a scale alignment module by minimizing the distances between scale distributions of source and domain scene. Wang et al. [25] propose to learn the domain shift at the parameter level and obtain the target model by a linear transformation. Gao et al. [26] propose multilevel feature aware adaptation (MFA) and structured density map alignment (SDA) to extract domain invariant features.

Adversarial training is also adopted to transfer the crowd counting model to a new scene. Wang et al. [14] leverage Cycle-Gan to translate synthetic data to surveillance images for crowd counting in real scenes. Following, the studies of [14,27] propose a domain adaptive method based on self-supervision without any manual label by translating synthetic data and generating pseudo labels on real scenes to improve the prediction quality. CODA [13] performs adversarial training with pyramid patches of multi-scales to deal with density distribution variations of source and target domain. Moreover, studies [28] also adopt the adversarial network to bridge the gap across domains considering local feature and crowd segmentation respectively.

The core of this method is to align the distribution of a source domain with a target domain, which is suitable for transferring to a specific new scene. However, in the multi-scene crowd density estimation problem, the model is often deployed in n unknown new scenes. If this kind of method is adopted, the deployers need to perform the model transfer for n times, which requires high labor costs.

#### 2.2.3. Meta Learning-Based Methods

In the study [10], the problem of deploying multi-scene counting methods in real-world surveillance applications for a new scene is presented, and finetune is suggested to reduce the cost of data annotation and model training. On this basis, considering fast adaptation to new target scenes, study [11] leverages Model-Agnostic Meta-Learning to learn the model parameters with strong generalization ability.

These methods obtain a model with strong generalization capability by learning the common knowledge in each scene. Taking advantage of the generalization, it takes a considerably small amount of labeled data and training steps to train a model for the new scene. Nevertheless, these methods still require model training for specific scenes. For a mass deployment of surveillance cameras (which is a common situation in practical applications), it is often necessary to train each scene individually to adapt the network model. Similar with domain adaptation-based methods, such adaption process requires the deployment staff to manually configure and train the neural network, with domain knowledge requirement and high labor cost.

### 2.3. Multi-Task Learning

First introduced in study [29], multi-task learning tries to promote the performance of multiple related tasks by exploiting the intrinsic relationships among them. It has been proven that by taking the similarity and difference of tasks into consideration, simultaneously learning the related tasks can achieve higher accuracy in solving similar tasks with different data distributions than merely pooling these samples to learn an overall model.

Based on the foundation that tasks are related via a certain structure, a lot of multi-task learning methods try to learn tasks with different sample distributions by trace-norm regularization [30], joint feature learning [31], shared hidden units in neural networks [29] and exploring tasks’ cluster tree and network structures [32,33].

Considering the outlier tasks in many real-world applications, study [34] extended the multi-task learning assumption that all tasks are related to each other and try to identify irrelevant (outlier) tasks, which is referred to as robust multi-task learning [35,36].

## 3. Methodology

In this section, we first describe the problem setup for multi-scene adaptive crowd counting and the pipeline of our proposed approach (Section 3.1). We then thoroughly illustrate the meta-knowledge in the field of crowd counting (Section 3.2) and the coarse two-stream density regression network proposed for meta-knowledge learning (Section 3.3). Finally, the fine-grained process based on multi-task learning is elaborated in Section 3.4.

### 3.1. Problem Setup and Proposed Pipeline

#### 3.1.1. Multi-Scene Adaption Crowd Counting

In a conventional supervised crowd density regression setting, there is a dataset $D_{all}=\left\{D_{}^{train},D_{}^{test}\right\}$, where $D^{train}$ and $D^{test}$ are training data and test dataset of all scenes respectively. The objective of crowd counting supervised methods is to learn a mapping $F\left(\theta \right)=X\to Y$, which can map the surveillance image X in $D_{}^{train}$ to its corresponding density map Y. Then, the overall counting of the image can be obtained by $C=\sum_{w=1}^{W}\sum_{h=1}^{H}Y_{w,h}$, where W and H is the width and height of density map Y.

Following study [11], we formulate the surveillance image crowd counting problem as a multi-task learning problem, where counting of each camera’s scene is considered as a specific task. We use $D_{i}=\left\{D_{i}^{train},D_{i}^{test}\right\}$ $i=\left(1,2,\cdots ,N\right)$ to denote the training and test dataset of the ith task $T_{i}$, where N denotes the total amount of known scenes with labeled counting images.

Different to the problem setting in study [11], we consider the problem of simultaneous deployment of multiple unknown scenes, which is more consistent with the real-world surveillance application. The counting task of unknown scenes is denoted as $T_{j}$, $j=\left(1,2,\cdots ,M\right)$. For each unknown scene, only K images are collected and labeled before the deployment of counting methods, which is denoted as $D_{j}=\left\{D_{j}^{train},D_{j}^{test}\right\}$. Thus, all the data of the N+M scenes can be denoted as $D_{all}=\left\{D_{1},\cdots ,D_{N},\cdots ,D_{N+M}\right\}$ in a multi- scene crowd counting problem.

#### 3.1.2. Overall Pipeline

Our goal of the proposed method is to learn an accurate counting model for multiple unseen scenes leveraging a few labeled and the existing labeled data of known scenes. Different from study [11], which train the model parameters $\theta_{meta}$ with meta learning and adapt it to each unknown scene to get $\theta =\left\{\theta_{1},\theta_{2},\cdots ,\theta_{M}\right\}$, this paper decomposes the objective into the following two sub objectives. As shown in Figure 1, the coarse objective is to learn an overall model containing scene-independent common knowledge with strong generalization to unseen scenes, while the fine-grained is to improve the accuracy by concentrating on the difference of multiple scenes including the unknown ones.

In our work, meta-knowledge learning is leveraged to promote the model $\theta_{all}$’s generalizability and performance by capturing useful scene-independent common knowledge. The overall model can be denoted as $F\left(\theta_{all}\right)=\left[X_{1},X_{2}\right]\to \left[Y_{1},Y_{2}\right]$, where $\left(\left[X_{1},X_{2}\right],\left[Y_{1},Y_{2}\right]\right)$ is sampled from labeled images of all scenes $D_{}^{train}$ and $\left[X_{1},X_{2}\right]$is image frame pair of the same scene.

Furthermore, we adopt the multi-task learning mechanism to explicitly concentrate on the difference of each scene and thus obtain the counting regression weight $w_{i}$ of multiple scenes simultaneously, including the unknown scenes. The regression process of each scene can be denoted as $f_{i}\left({\hat{Y}}_{i}\right)=w_{i}^{T}{\hat{Y}}_{i}\to C_{i}$.

### 3.2. Meta-Knowledge Analysis

As the data distribution of new scenes is uncertain, it is crucial to learn common meta-knowledge from the labeled data of existing scenes to achieve generalization to new ones. For crowd density estimation, the meta-knowledge can be summarized into the following categories: foreground/background knowledge, perspective scale knowledge and inter-frame knowledge. Among them, foreground/background knowledge and perspective scale knowledge mainly belong to the spatial domain of a single image, while inter-frame knowledge belongs to the temporal domain and is obtained by comparing multiple images of the same camera scene.

#### 3.2.1. Foreground/Background Knowledge

In surveillance videos, the crowd foreground shares similar characteristics in different scenes. Such foreground knowledge enables the most basic crowd density regression networks to have a fundamental counting capacity. Thus, basic generalization capability is possessed by common crowd density estimation methods.

As for the background, where there are few crowd distributions, often showing trees, buildings, pools, roads, etc., such common objects of backgrounds show similarities on different scenes. The reason why some methods incorporating background segmentation can improve counting performance is that they explicitly improve the learning ability on background knowledge.

#### 3.2.2. Perspective Scale Knowledge

This knowledge, which belongs to the category of single pictures, includes both common knowledge of all scenes and specific knowledge of each scene. The common knowledge of the perspective scale that exists in all scenes is manifested by the fact that the near pedestrian target is larger, while the target scale becomes smaller as the distance from the camera becomes farther. The scene-specific knowledge, caused by different surveillance camera locations, is the differences in the crowd scale variation range of surveillance video frames in different scenes. The scale variation of targets is a research hotspot in the field of crowd counting. Many network structures have been proposed in existing studies to solve this problem, significantly facilitating the learning of this knowledge and thus promoting counting accuracy.

#### 3.2.3. Inter-Frame Temporal Knowledge

This knowledge is a potential pattern implied between sequential video frames. By comparing and analyzing two or more images, changes tend to be the foreground crowds, while the unchanged areas have a high probability of being the background. In the existing studies of crowd counting, researchers tend to focus on the knowledge of the crowd, background and scale variation in a single image, while ignoring this knowledge between frames of the same camera in multi-scene surveillance applications. In addition to the knowledge of single image, the inter-image knowledge helps to assist model reasoning in ambiguous situations. For example, dense foliage is similar to the distribution of a distant crowd. Such similarity may confuse the model if it is solely based on the feature of a single image. However, as crowds are always moving, it is easier to distinguish the stationary background from the foreground crowd by comparing inter-frame images. As a result, such knowledge reduces interference of background and thus promotes the sensing of the foreground crowd.

### 3.3. Two-Stream Network Structure

Most of the existing crowd density estimation methods based on CNN adopt a single-stream network structure which mainly consists of the encoder-decoder, as shown in Figure 2. The encoder accepts picture X as the network input and gets feature F. The decoder maps the feature F to the crowd density map Y. According to the analysis in Section 3.2, such a structure can learn the spatial domain knowledge of a single image, including foreground crowd knowledge, background knowledge and perspective scale knowledge, without the inter-frame temporal knowledge of surveillance videos.

A study [37] has indicated that estimation errors in the background areas impede the performance of the counting methods. To address this problem, some existing methods [18,28] add crowd segmentation branches to crowd density regression, which improves the ability of the network to perceive and segment the background. However, existing studies are based on a single image and ignore the important domain knowledge that the background of the video frame in the same scene is consistent. Inter-frame temporal knowledge can improve the crowd segmentation effectively in each scene. In this paper, a two-stream network structure is used to learn this meta-knowledge.

The comparison of K (K≥2) images is needed to better perceive inter-frame temporal information. In order to reduce the complexity of the model, an intuitive two-stream neural network (K=2) was adopted to learn this knowledge by comparing the high-level features obtained by the encoder. As illustrated in Figure 2, the two-stream network adopted a Siamese mechanism to reduce parameters, with two video frames $X_{1}$ and $X_{2}$ of the same scene as network inputs and the encoder obtained features $F_{1}$ and $F_{2}$, respectively.

After differencing $F_{1}$ and $F_{2}$, they were respectively concatenated with the original high-level features to obtain features $F_{1,C}$ and $F_{2,C}$, which is shown as follows:$$F_{1,C}=F_{1}⊕\left(F_{1}-F_{2}\right)\;F_{2,C}=F_{2}⊕\left(F_{2}-F_{1}\right)$$

As inputs to the decoder, $F_{1,C}$ and $F_{2,C}$ mapped the differential fusion features to the estimated crowd density maps ${\hat{Y}}_{1}$ and ${\hat{Y}}_{2}$, respectively, and trained the neural network with the density of ground truth obtained by calculation.

The motivation of difference mainly lies in highlighting the changing and stable areas by comparison. The stable areas of two images in the same scene are more likely to be the background. This network structure is capable of learning meta-knowledge in both the time and space domains.

**Backbone network architecture**. Our proposed two-stream network can apply any crowd counting network structure with encoder and decoder, which is the mainstream crowd counting network structure. In this paper, we use CSRnet [38] as our backbone, as it has been proved to achieve favorable performance with a simple and elegant structure. The network consists of an encoder as feature extractor and a decoder as density map estimator. The encoder makes use of VGG-16 [39] to extract the feature of the input image. The decoder consists of a series of dilated convolutional layers, which are used to regress the output density map.

### 3.4. Counting Regression Based on Multi-Task Learning

Most existing CNN-based counting methods try to count people by direct integration of the estimated density maps which are the output of the overall model trained with pooling samples of all scenes [3,6,40,41]. As a result, the error in estimated density maps, caused by the fact that the overall network ignores the difference among scenes, directly accumulates to errors in the final crowd counting.

Our aim is to promote the accuracy of overall crowd counting results in multiple scenes. To this end, instead of the direct integration applied by existing methods, we propose to adopt an additional learning process to project density maps to overall counts. More specifically, we propose to adopt the multi-task learning method for multiple scenes.

The training samples of w scenes are composed of feature vectors reshaped from the estimated density maps and the overall counting acted as the regression label. We use $\left(X_{i,k},{\hat{Y}}_{i,k},C_{i,k}\right)$ to denote the labeled image samples, where i is the index of the scene and *k* is the index of the *k*th frame image in this scene. ${\hat{Y}}_{i,k}$ is the estimated density map of the kth training sample of the ith scene and $C_{i,k}$ is the number of pedestrians in frame $X_{i,k}$.

Based on the assumption that the previous density map regression network has already finished the non-linear projection from the surveillance frames to density maps, a linear function $f_{i}$ is capable to regress the density map to overall counting.
$$f_{i}\left({\hat{Y}}_{i,k}\right)=w_{i}^{T}{\hat{Y}}_{i,k}\approx C_{i,k}$$
where $w_{i}$ denotes the linear regression weight of the ith scene.

Without considering the robust multi-task learning penalty, the crowd counting regression problem can be formulated as
$$\min_{W}\;\underset{i=1}{\sum^{N}}\;\underset{k=1}{\sum^{K_{i}}}\;\frac{1}{\sum_{i=1}^{N}K_{i}}{∥w_{i}^{T}{\hat{Y}}_{i,k}-C_{i,k}∥}^{2}$$
where $K_{i}$ is the amount of labeled frames in the ith scene.

In real-world crowd counting applications, the background and person distribution of various scenes may vary on a large scale. Thus, the existence of *outlier* scenes is inevitable and may probably mislead the model in other tasks if not properly dealt with. To promote the robustness of the multi-task counting regression in applications, we adopt the method proposed by [34], where the regression tasks in various scenes are divided into two groups: the related scenes group and the *outlier* scenes group.

Inspired by study [35], we adopted a low-rank structure to couple the *closely-related* scenes and utilize a group-sparse structure to identify *outlier* scenes. Specifically, the weight of the regression model in ith scene can be denoted as
$$w_{i}=l_{i}+s_{i}$$
where $l_{i}$ and $s_{i}$ are the low-rank and the group-sparse structure of the weight $w_{i}$, respectively. Such decomposition is based on the motivation that the ith task should be either an *outlier* task or a *closely-related* task. If the ith task is from the *closely-related* tasks group, $s_{i}$ is expected to be a zero-vector and hence $w_{i}$ obeys the specified low-rank structure constraint. on the other hand, if the ith task is from the *outlier* tasks group, $s_{i}$ is expected to be non-zero and $w_{i}$ is equal to the direct value of $l_{i}$ plus the non-zero $s_{i}$.

As the regression matrix of the m scenes can be denoted by W=w1,⋯wN,W∈ℝWH×N, the weight matrix W can be decomposed to two components, namely, low-rank matrix L=l1,⋯,lm and group-sparse matrix S=s1,⋯,sm.

To achieve this intuition, on the one hand, we adopt the trace norm regularization term on L to encourage the low-rank structure. On the other hand, the $l_{1,2}$-norm regularization term is used to induce group sparse structure in the matrix S. Then the robust multi-scene crowd counting regression problem can be formulated as
$$\min_{W}\;\underset{i=1}{\sum^{N}}\;\underset{k=1}{\sum^{K_{i}}}\;\frac{1}{\sum_{i=1}^{N}K_{i}}{∥w_{i}^{T}{\hat{Y}}_{i,k}-C_{i,k}∥}^{2}+\alpha \left\|L{\|}_{*}+\beta \right\|S{\|}_{1,2}$$
where *α* and *β* are non-negative parameters.

Note that similar robust multi-task learning formulation based on low-rank and group-sparse structures are studied in [35,36], which focus on the accelerated proximal method to solve the problem and provide the performance bound of the formulation. Here we adopt the solving method proposed by [35] to solve the robust multi-task formulation.

The previous two-stream crowd density estimation network can be regarded as a sharing feature extractor for all scenes and the density maps can be considered as features for each image. The fine-grained multi-task learning process can also be seen as another specific layer, whose parameters vary to suit different scenes and obey the multi-task penalty.

## 4. Experiments

In this section, we first introduce the evaluation metrics and datasets (Section 4.1). We then describe several baselines for comparison and the experiment setup (Section 4.2). The experimental results are presented afterward (Section 4.3 and Section 4.4).

### 4.1. Metrics and Dataset

#### 4.1.1. Evaluation Metrics

In this paper, we adopt the commonly used metrics in the field of crowd count estimation, namely mean absolute error (*MAE*), root mean squared error (*MSE*) and mean deviation error (*MDE*) in [3,42] to evaluate the performance of each method.
$$MAE=\frac{1}{\sum_{i=1}^{M}K_{i}}\sum_{i-1}^{N}\sum_{k=1}^{K_{i}^{test}}\left|C_{i,k}-{\hat{C}}_{i,k}\right|\;MDE=\frac{1}{\sum_{i=1}^{M}K_{i}}\sum_{i=1}^{M}\sum_{k=1}^{K_{i}^{test}}\frac{\left|C_{i,k}-{\hat{C}}_{i,k}\right|}{C_{i,k}}$$
$$RMSE=\sqrt{\frac{1}{\sum_{i=1}^{M}K_{i}}\sum_{i-1}^{N}\sum_{k=1}^{K_{i}^{test}}{\left|C_{i,k}-{\hat{C}}_{i,k}\right|}^{2}}$$
$$MDE=\frac{1}{\sum_{i=1}^{M}K_{i}}\sum_{i=1}^{M}\sum_{k=1}^{K_{i}^{test}}\frac{\left|C_{i,k}-{\hat{C}}_{i,k}\right|}{C_{i,k}}$$
where M is the number of new scenes and $K_{i}^{test}$ is the number of test images in the ith surveillance scene. Moreover, $C_{i,k}$ and ${\hat{C}}_{i,k}$ denote the ground truth and estimated overall counting, respectively. ${\hat{C}}_{i,k}$ can be calculated by multi-task regression model as illustrated in Section 3.4.

#### 4.1.2. Dataset

Many available datasets in the context of crowd counting neglect the multi-scene domain knowledge in surveillance applications. As a result, they just randomly collect and pool the crowd images of arbitrary scenes. Different from arbitrary image crowd counting methods, the multi-scene adaptive problem formulated in this paper should be evaluated on datasets containing the scene information. More specifically, the video frames should be collected by multiple fixed cameras and each collected frame should be annotated, not only the location of each person, but also the camera ID as well.

To the best of our knowledge, WorldExpo10 dataset [42] is the only large-scale multi-scene dataset. It contains 1132 images collected from 108 different surveillance cameras on the campus of WorldExpo10. The dataset is released in the cross-scene application, where annotated images of 103 scenes are used as training data and the remaining images of 5 scenes as test data. Following study [42], in the multi-scene adaptive crowd counting application, we randomly set 103 known scenes and 5 unseen scenes with few labeled images. The labeled data in known scenes is used as training data for learning the generic model at the coarse-grained stage. For unknown scenes, K=5 labeled images in each scene are adopted to train the multi-task regression weights at the fine-grained stage.

### 4.2. Baselines and Implementation

#### 4.2.1. Baselines

To compare the performance on multi-scene adaptive crowd counting, we adopt methods of three mainstream domain adaption crowd counting pipelines as baseline.

Fully supervision methods for arbitrary scenes: Cross-scene net [42], CSRnet [38], CAN [41] and AMSNet [43].Domain adaption-based crowd counting: SE Cycle GAN [14]. Through adversarial training-based domain adaption, the labeled data of known scenes are pooled together and thus regarded as source domain, while each unseen scene is regarded as target domain. By implementing the domain adaption training for 5 scenes successively, SE Cycle GAN learns 5 models specifically tuned to each particular scene.Meta learning-based adaptive crowd counting: MAML-counting [11]. MAML mechanism is leveraged to learn the model parameters of fast adaptation to target scenes. Similar with SE Cycle GAN, models suitable for each scene are fine-tuned respectively.

#### 4.2.2. Implementation Details

(1) Crowd density regression. In the process of learning meta knowledge, the ground truth of overall model is the density map calculated from the dot annotations given by dataset. Following conventional mechanism in the field of crowd counting [3,41], we adopt a Gaussian kernel to blur the point annotations. Similar with backbone, the proposed two-stream network’s (TSN) encoder is initialized with the weights of pre-trained VGG [39]. We set batch size to be 8 with Adam optimizer [44] adopted for parameter update. The TSN is trained for 500 epochs. For each epoch, 8 pairs of images are randomly sampled from the labeled images of each scene. Such image pair also contributes to data augmentation, which makes the labeled training samples of the ith scene increasing from $K_{i}$ to $K_{i}\cdot (K_{i}-1)/2$, where $K_{i}$ is the number of labeled training images of the ith scene.

(2) Multi-task learning: We apply the MALSAR toolbox to solve the robust multi-task regression problem. The maximum number of iterations is set as 1500. The parameter of low rank regularization is α=10, while that of group-sparse is β=30.

### 4.3. Comparison on Performance

To verify the performance of the proposed method, we perform comparison experiments of the existing methods as introduced in Section 4.2.1. As shown in Table 1, the performance of fully supervision methods becomes better from Cross Scene Net to AMSNet as the network structure becomes more and more complex. Without considering the specific characteristic of scenes, AMSNet still achieves acceptable performance, which demonstrates the effect of foreground/background common knowledge and perspective scale knowledge. SE Cycle GAN and MAML-counting are domain adaption methods, whose model training processes are dependent on each specific target domain. They show better performance compared with their backbone, which indicates that scene-specific model can promote counting accuracy. Our proposed method TSN with multi-task learning (MTL) achieves dominant counting accuracy with an overall MAE of 6.1, outperforming the representative methods of two kinds, namely AMSNet and MAML-counting by 10.29% and 13.48%, respectively.

To analyze the improvement of performance by MTL at fine-grained stage, we compare the effect of coarse-grained crowd counting, which only adopts TSN with the performance of the proposed TSN with MTL method. Compared with the TSN method without multi-task fine-grained stage, the proposed method can reduce the average MAE by 15.28%. Moreover, Backbone (CRSNet) with MTL can reduce the MAE by 12.79%. Such improvement embraced by MTL demonstrates its universal optimization effect on accuracy in multi-scene crowd counting problem. Further discussion about the effect of MTL is illustrated in Section 4.4.

To discuss the performance of TSN, we compare it with its backbone. Compared with the CRSNet, the proposed TSN reduces MAE by 16.28%, which demonstrates the capacity of inter-frame temporal knowledge. Note that our TSN is a generic model for domain generalization such as fully supervised methods. Moreover, backbone (CRSnet) with MTL means that it does not use a meta-learning mechanism and merely adopts a single stream network to estimate single image’s density map, and then carries out fine-grained stage with multi-task learning. Compared with backbone (CRSnet) with MTL, the proposed TSN with MTL method can reduce the average MAE by 18.67%. In addition, we provide visual results of the backbone CRSNet and the proposed TSN in Figure 3. Figure 3c shows that the error on background areas is a difficult issue to be addressed in counting. From the comparison of Figure 3c,d, the STN shows its capacity on recognizing background, which comes from the learning of meta inter-frame knowledge.

To further discuss the robustness of the algorithm under various conditions with different lighting and crowd density, we select surveillance images collected from cameras, whose IDs are 100,400 and 100,730, as shown in Figure 4 and Table 2. In WorldExpo’10, camera 100,400 collects images of different illumination conditions, while camera 100,730 faces obvious fluctuation of crowds. By observing the estimated density map corresponding to 100,730, it can be found that although the crowd density significantly varies, the density maps obtained by the proposed TSN and TSN with MTL method are accurate and clear, with few noises in background areas, which shows that the proposed method is capable to distinguish human crowd with buildings and trees in the background. Therefore, the proposed method has favorable robustness in the case of large population density divergence. The main reason can be attributed to the STN’s capacity on learning the knowledge to adopt the difference between frames to distinguish the foreground crowd from the background.

The proposed method shows relatively poor performance in the scene of camera 100,400. Such performance degradation can be observed in the noise of the crowd density maps and the metrics in Figure 2. The TSN may find it difficult to distinguish the difference caused by lighting from the difference caused by foreground crowd movement, thus confusing the recognition of crowds. Such confusion may be caused by the unbalanced data distribution of video frames. More specifically, most of the video frames in this data set are collected during the day, while less than 10% of frames are collected at night. Moreover, the proposed method has not elaborated a data sampling mechanism to balance the data distribution of various illumination conditions. Therefore, only a small part of picture pairs sent into the STN are under different lighting conditions. Improving the sampling mechanism to balance the lighting conditions of image pairs, so as to optimize the counting robustness under different lighting conditions, is a research direction worthy of further research.

### 4.4. Multi-Task Learning

The necessity of the fine-grained stage using multi-task learning results from the difference of data distribution collected from various scenes. To evaluate the necessity, we adopt the regression parameters of each scene which is learned in multi-task learning process to represent the data distribution and explore the similarity of multiple scenes. Cosine similarity is adopted to calculate the similarity matrix M of 108 scenes, shown in Figure 4; the similarity between ith and jth scenes is denoted by $M_{i,j}$ and calculated by
$$M_{ij}=\frac{\sum_{k=1}\;w_{k}^{i}w_{k}^{j}}{\sqrt{\sum_{m=1}\;{\left(w_{m}^{i}\right)}^{2}}\sqrt{\sum_{n=1}\;{\left(w_{n}^{j}\right)}^{2}}}$$
where $w_{i}$ is the regression parameter of the ith scene.

In Figure 5, the ith and jth scene are more similar if $m_{i,j}$ is brighter in Figure 5. It is clear that 72–80 scenes are different with 1–71 scenes and 10–100 scenes. We realize that the 108 scenes can be generally divided into four groups which in turn illustrated the necessity to induce the multi-task learning method to explore the difference and similarities of multiple scenes and learn a set of regression parameters for all scenes.

More specifically, the difference of scenes, which is presented by the similarity matrix in Figure 5, demonstrates that merely adopting the overall CNN-based density map estimation with direct integration mechanism can barely capture the specific data distribution of each scene. The direct integration of the estimated density map can be expressed as ${\hat{C}}_{i,k}=\sum {\hat{Y}}_{i,k}$, where ${\hat{Y}}_{i,k}$ is the estimated density map of the kth training sample of the ith scene and ${\hat{C}}_{i,k}$ is the estimated overall crowd counting. Compared with the direct integration mechanism adopted by conventional CNN-based crowd counting methods, the proposed MTL-based fine-grained process learns the regression weight from the estimated density map to the overall counting. With the specific learned regression weight of each scene, the overall counting can be calculated as weighted integration of the regression parameters and the estimated density map, ${\hat{C}}_{i,k}=w_{i}^{T}{\hat{Y}}_{i,k}$, where $w_{i}$ is the regression weight of the ith scene. Such weighted integration captures the character of each scene in addition to the overall CNN-based density map estimation process.

On the other hand, the similarity of scenes indicates that the density map regression tasks of the known scenes with sufficient labeled images can promote the regression performance of unknown new scenes with few labeled images, which can largely reduce the demand of training samples when cameras need to be deployed in new locations. The data of other labeled similar scenes can facilitate the counting regression of new scenes with the advantage of MTL mechanism.

In all, due to the differences and similarities between scenes, the MTL-based density map regression can promote overall counting accuracy and reduce the demand for labeled images of new scenes, respectively. Such weighted integration using the learned regression parameters can also be regarded as a type of attention mechanism, with MTL regularization items added.

Such relation of the multiple tasks is the reason why the fine-grained stage using multi-task learning can promote the counting accuracy in Table 1. Compared with TSN’s performance, TSN with MTL can reduce the MAE by 15.28%.

## 5. Conclusions

This paper pioneers the analysis of the crowd density estimation problem from a knowledge learning perspective. In this paper, a coarse-to-fine pipeline is adopted to solve the multi-scene adaptive problem in crowd counting. The objective of coarse stage is to learn a generic model robust to unseen scenarios. We observe that meta-knowledge (i.e., scene-independent common knowledge) is the cornerstone for generalization capacity and analyze the meta-knowledge of crowd counting. Exploiting the inter-frame temporal knowledge, a two-stream network structure is adopted to optimize the perception of foreground crowds and promote generalization ability to unknown scenes. In addition, at the fine-grained stage, a robust multi-task method is adopted to train the counting regression parameters of each specific scene, and thus promote the counting accuracy in several new scenes simultaneously.

However, we also note that the proposed method is a two-phase approach. In future work, the overall counting regression from density map can essentially be replaced by a specific layer of the neural network. By transforming it into a holistic model, we will explore an end-to-end network that integrates domain adaptation with meta-knowledge learning.

## Figures and Tables

**Figure 1 sensors-22-03320-f001:**
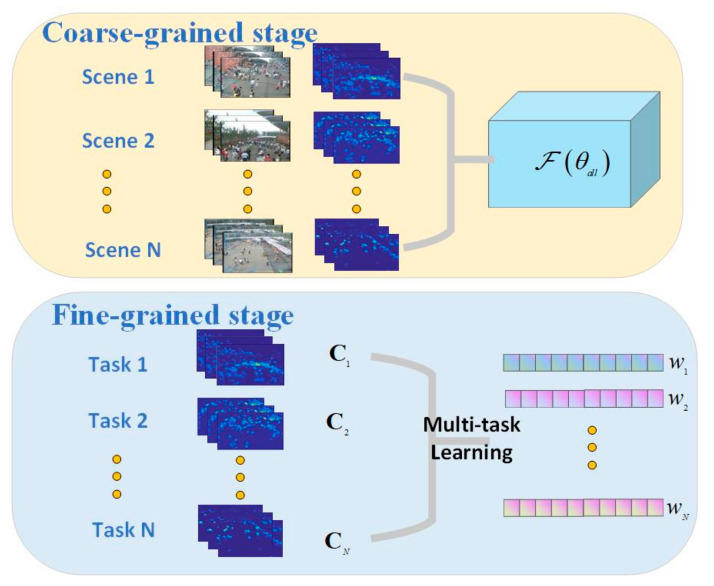
The pipeline of proposed coarse-to-fine multi-scene adaptive crowd counting. At the coarse-grained stage, the frame pairs of multiple known scenes are used to train a generic model with meta-knowledge. At the fine-grained stage, overall counting regression from estimated density maps of each scene is regarded as a specific task. Multi-task learning is used to learn the regression weight of each specific scene.

**Figure 2 sensors-22-03320-f002:**
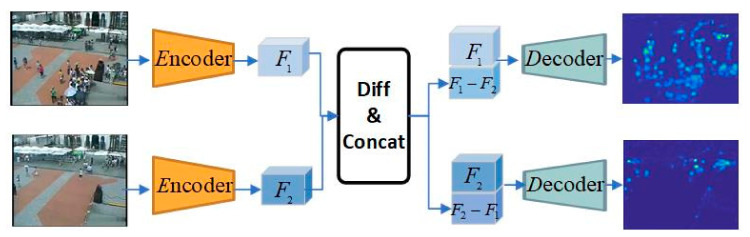
Two-stream network to capture meta-knowledge.

**Figure 3 sensors-22-03320-f003:**
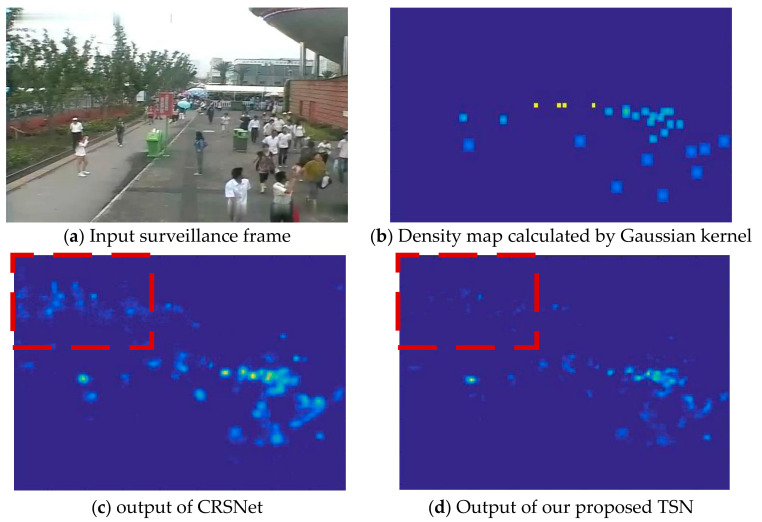
Comparison of density maps estimated by CRSNet and our proposed TSN.

**Figure 4 sensors-22-03320-f004:**
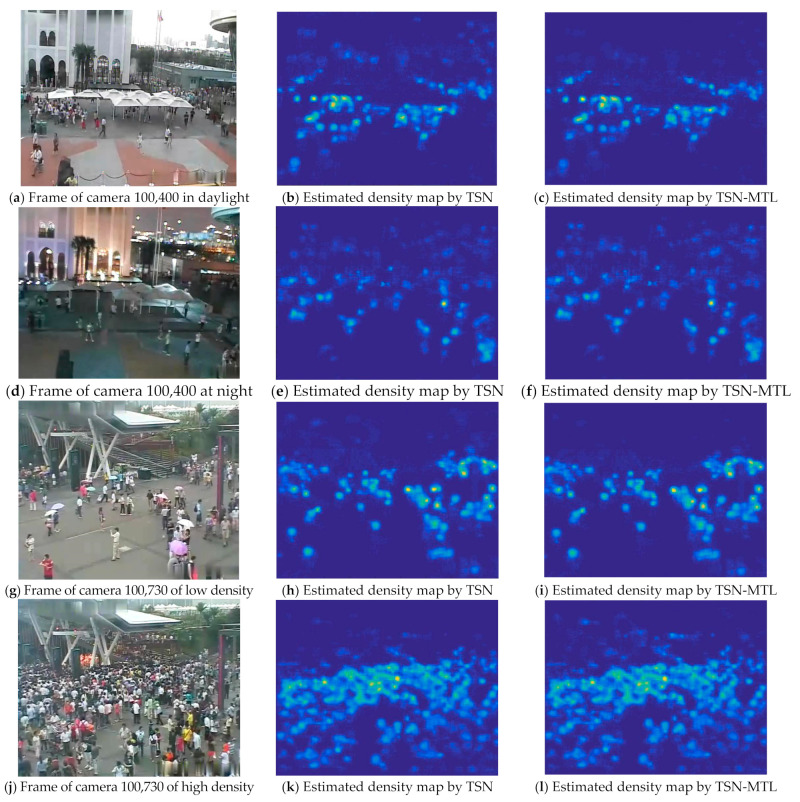
Robustness performance of the proposed TSN-MTL under different lighting conditions and crowd density. Display of estimated crowd density map of pictures collected by camera 100,400 and 100,730. The first column is the original picture, the second column is the density map estimated by STN and the third column is the density map estimated by STN with MTL. The first two rows are frames collected by camera 100,400, while the next two rows are frames collected by camera 100,730.

**Figure 5 sensors-22-03320-f005:**
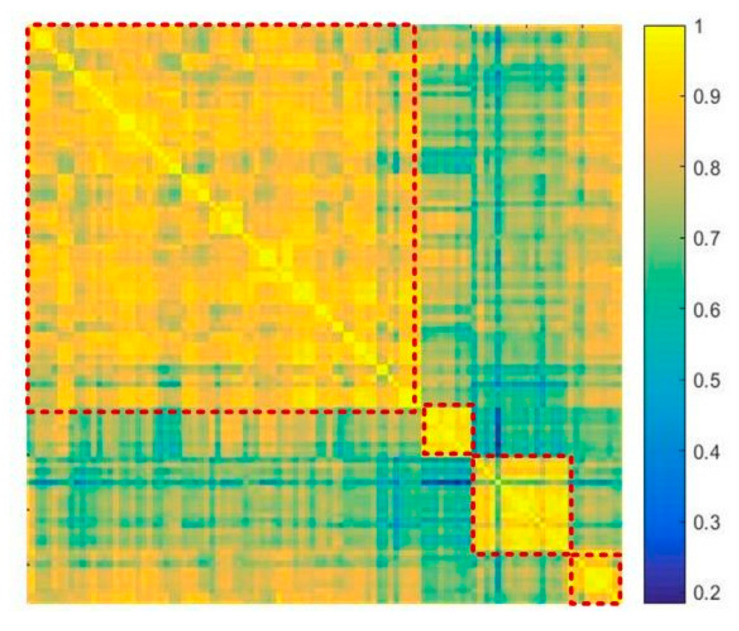
Similarity relationship of parameters in multiple scenes.

**Table 1 sensors-22-03320-t001:** The MAE comparison on WorldExpo’10. The best performance is colored red and second best is colored blue.

Methods		S1	S2	S3	S4	S5	Ave.
Fully Supervised Methods	Cross Scene Net	9.8	14.1	14.3	22.2	3.7	12.9
	CRSNet	2.9	11.5	8.6	16.6	3.4	8.6
	CAN	2.9	12.0	10.0	**7.9**	4.3	7.4
	AMSNet	**1.6**	**8.8**	10.8	10.4	**2.5**	**6.8**
Domain Adaption Methods	SE Cycle GAN	2.7	15.4	12.1	11.9	3.6	9.1
	MAML-counting	3.05	10.37	8.18	9.41	3.91	7.05
Proposed Methods	TSN	2.8	9.2	8.9	11.7	3.2	7.2
	TSN with fine-tune	2.5	9.0	**8.1**	11.3	**2.8**	6.8
	Backbone (CRSNet) with MTL	2.5	10.3	8.4	13.4	3.0	7.5
	Proposed TSN with MTL	**2.2**	**8.7**	**7.6**	**9.3**	**2.5**	**6.1**

**Table 2 sensors-22-03320-t002:** The performance comparison on camera 100,400 and camera 100,730.

Methods	100,400			100,730		
	*MAE*	*RMSE*	*MDE* (%)	*MAE*	*RMSE*	*MDE* (%)
TSN Proposed TSN with MTL	11.56	16.35	34.96	7.44	11.57	10.57
	7.98	12.06	20.49	5.01	8.83	6.17

## Data Availability

Not applicable.

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
