# Peer review of "Meta-Knowledge and Multi-Task Learning-Based Multi-Scene Adaptive Crowd Counting"

_sensors, 2022, doi:10.3390/s22093320_

Round 1
Reviewer 1 Report
This paper proposes a multi-scene adaptive crowd counting method based on meta-knowledge and multi-task learning. The proposed method outperforms the baselines in most cases. The writing is professional and clear. However, there should be some ablation studies to demonstrate the effectiveness of each component and design. For example, what will happen when we remove meta-knowledge learning or multi-task learning? What will happen when we use a single-stream network structure? What will happen when remove the coarse-grained stage or the fine-grained stage? Moreover, some important references are missing, like “DecideNet: Counting Varying Density Crowds Through Attention Guided Detection and Density” (CVPR 2018), “Crowd Counting With Deep Negative Correlation Learning” (CVPR 2018), and “Scale Aggregation Network for Accurate and Efficient Crowd Counting” (ECCV 2018).
Author Response
We sincerely thank the editor and all reviewers for their valuable feedback that we have used to improve the quality of our manuscript. The reviewer comments are laid out below in italicized font and specific concerns have been numbered. Our response is given in normal font and changes/additions to the manuscript are given in red text.
- there should be some ablation studies to demonstrate the effectiveness of each component and design. For example, what will happen when we remove meta-knowledge learning or multi-task learning? What will happen when we use a single-stream network structure? What will happen when remove the coarse-grained stage or the fine-grained stage?
The following two paragraph is added in Section 4.3
To analyze the improvement of performance by MTL at fine-grained stage, we compare the effect of coarse-grained crowd counting, which only adopts TSN with the performance of the proposed TSN with MTL method. Compared with the TSN method without multi-task fine-grained stage, the proposed method can reduce the average MAE by 15.28%. Moreover, Backbone (CRSNet) with MTL can reduce the MAE by 12.79%. Such improvement embraced by MTL demonstrate its universal optimization effect on accuracy in multi-scene crowd counting problem.Further discussion about the effect of MTL is illustrated in Sec 4.4.
To discuss the performance of TSN, we compare it with its backbone. Compared with the CRSNet, the proposed TSN reduces MAE by 16.28%, which demonstrates the capacity of inter-frame temporal knowledge. Note that our TSN is a generic model for domain generalization like fully supervised methods. Moreover, backbone (CRSnet) with MTL means that it does not use meta learning mechanism and merely adopts single stream network to estimate single image's density map, and then carries out fine-grained stage with multi task learning. Compared with backbone (CRSnet) with MTL, the proposed TSN with MTL method can reduce the average MAE by 18.67%.
- “Moreover, some important references are missing, like “DecideNet: Counting Varying Density Crowds Through Attention Guided Detection and Density” (CVPR 2018), “Crowd Counting With Deep Negative Correlation Learning” (CVPR 2018), and “Scale Aggregation Network for Accurate and Efficient Crowd Counting” (ECCV 2018).”
The following paragraph introducing the important crowd counting methods is added in page 3 line 124-131
Cao et al. [17] adopt scale aggregation modules to extract multi-scale features and propose a novel training loss combining Euclidean loss and local pattern consistency loss. To improve generalization capability, Shi et al. propose decorrelated ConvNet[18], where a pool of decorrelated regressors are trained. Considering the fact that detection mechanism is more suitable to low density scenes, while regression is more applicable for congested areas. Liu et al. propose DecideNet[19], which can adaptively decide whether to adopt regression pipeline or detection pipeline for different locations based on its density conditions

Reviewer 2 Report
In the paper I consider that several examples should be presented to confirm the performance of the algorithm. The reader must understand the performance of the proposed solution for different situations such as images from the same location with different number of people present in the frame or different lighting situations.
Author Response
We sincerely thank the editor and all reviewers for their valuable feedback that we have used to improve the quality of our manuscript. The reviewer comments are laid out below in italicized font and specific concerns have been numbered. Our response is given in normal font and changes/additions to the manuscript are given in blue text.
- In the paper I consider that several examples should be presented to confirm the performance of the algorithm. The reader must understand the performance of the proposed solution for different situations such as images from the same location with different number of people present in the frame or different lighting situations.
Thanks to the suggestions of experts, we deeply examined the robustness of the method with different crowd density and lighting conditions. The robustness of the proposed method needs to be improved. In the future research, we will try to improve the sampling mechanism of image pairs to make the change of illumination more balanced among training samples, which may enable the neural network to distinguish the pixel change caused by illumination from the change caused by crowd movement. Such knowledge may reduce the error of crowd facing various illumination
The following examples and analysis are added in Section 4.3.
Figure 4. Display of estimated crowd density map of pictures collected by camera 100400 and 100730. The first column is the original picture, the second column is the density map estimated by STN, and the third column is the density map estimated by STN with MTL. The first two rows are frames collected by camera 100400, while the next two rows are frames collected by camera 100730.
Table 2. The performance comparison on camera 100400 and camera 100730.
|
METHODS |
100400 |
100730 |
||||
|
MAE |
RMSE |
MDE(%) |
MAE |
RMSE |
MDE(%) |
|
|
TSN Proposed TSN with MTL |
11.56 |
16.35 |
34.96 |
7.44 |
11.57 |
10.57 |
|
7.98 |
12.06 |
20.49 |
5.01 |
8.83 |
6.17 |
|
To further discuss the robustness of the algorithm under various conditions with different lighting and crowd density, we select surveillance images collected from cameras, whose IDs are 100400 and 100730, as shown in Fig. 4 and Tab 2. In WorldExpo10, camera 100400 collects images of different illumination conditions, while camera 100730 faces obvious fluctuation of crowds. By observing the estimated density map corresponding to 100730, it can be found that although the crowd density significantly varies, the density maps obtained by the proposed TSN and TSN with MTL method are accurate and clear, with few noises in background areas, which shows that the proposed method is capable to distinguish human crowd with buildings and trees in the background. Therefore, the proposed method has favorable robustness in the case of large population density divergence. The main reason can be attributed to the STN’s capacity on learning the knowledge to adopt the difference between frames to distinguish the foreground crowd from the background.
The proposed method shows relatively poor performance in the scene of camera 100400. Such performance degradation can be observed in the noise of the crowd density maps and the metrics in Figure 2. The TSN may find it difficult to distinguish the difference caused by lighting from the difference caused by foreground crowd movement, thus confusing the recognition of crowds. Such confusion may be caused by the unbalanced data distribution of video frames. More specifically, most of the video frames in this data set are collected during the day, while less than 10% of frames are collected at night. Moreover, the proposed method has not elaborated a data sampling mechanism to balance the data distribution of various illumination conditions. Therefore, only a small part of picture pairs sent into the STN are under different lighting conditions. Improving the sampling mechanism to balance the lighting conditions of image pairs, so as to optimize the counting robustness under different lighting conditions is a research direction worthy of further research.

Round 2
Reviewer 1 Report
The revised version is ready to be published.
Author Response
We sincerely thank the editor and all reviewers for their valuable feedback that we have used to improve the quality of our manuscript.